# Tracing Pharmacological Knowledge in Large Language Models

**Basil Hasan Khwaja**
School of Electrical and Computer Engineering
Purdue University
West Lafayette, IN, USA
bkhwaja@purdue.edu

**Dylan Chen**
Viterbi School of Engineering
University of Southern California
Los Angeles, CA, USA
dylanach@usc.edu

**Guntas Toor**
Stephen J. R. Smith Faculty of Engineering and Applied Science
Queen's University
Kingston, ON, Canada
guntas.toor@queensu.ca

**Anastasiya Kuznetsova**
Department of Molecular and Cellular Biology
Scripps Research
La Jolla, CA, USA
akuznetsova@scripps.edu

## Abstract

Large language models (LLMs) have shown strong empirical performance across pharmacology and drug discovery tasks, yet the internal mechanisms by which they encode pharmacological knowledge remain poorly understood. In this work, we investigate how drug-group semantics are represented and retrieved within Llama-based biomedical language models using causal and probing-based interpretability methods. We apply activation patching to localize where drug-group information is stored across model layers and token positions, and complement this analysis with linear probes trained on token-level and sum-pooled activations. Our results demonstrate that early layers play a key role in encoding drug-group knowledge, with the strongest causal effects arising from intermediate tokens within the drug-group span rather than the final drug-group token. Linear probing further reveals that pharmacological semantics are distributed across tokens and are already present in the embedding space, with token-level probes performing near chance while sum-pooled representations achieve maximal accuracy. Together, these findings suggest that drug-group semantics in LLMs are not localized to single tokens but instead arise from distributed representations. This study provides the first systematic mechanistic analysis of pharmacological knowledge in LLMs, offering insights into how biomedical semantics are encoded in large language models.

## Meaningfulness Statement

A meaningful representation of life captures biologically grounded concepts in a structured and interpretable form. In biomedical LLMs, this entails encoding pharmacological knowledge that can be localized and intervened. Our work advances this goal by mechanistically characterizing how drug groups are represented in Llama-based models, identifying early-layer representations that are causally predictive of drug-group associations. We further show that pharmacological semantics are distributed across tokens and are stable across layers.

# 1 INTRODUCTION

Large language models (LLMs) have recently emerged as a powerful foundation for a wide range of tasks in pharmacology Trajanov et al. (2022) and drug discovery Lu et al. (2025), driven by their ability to model complex biomedical knowledge from large-scale text corpora. LLMs have been applied to target identification Liu et al. (2021), drug–drug interaction prediction Sicard et al. (2025), and automated hypothesis generation Younis et al. (2025), particularly when deployed within agentic frameworks. Recent work further demonstrates that LLMs can be integrated with structured resources such as knowledge graphs and biomedical databases to support drug repurposing Wang et al. (2025) and literature-driven drug discovery chin Huang et al. (2025).

However, despite promising empirical results, the internal mechanisms by which LLMs encode pharmacological concepts, such as drug classes, functional groups, or therapeutic actions, remain poorly understood. This has motivated growing interest in mechanistic interpretability and representation analysis of biomedical LLMs, with the goal of localizing where and how pharmacological semantics emerge across layers and components of the model. Such understanding is critical not only for improving model reliability and generalization, but also for establishing LLMs as scientifically trustworthy tools in high-stakes biomedical domains.

In this work, we use activation patching to causally demonstrate that early layers in Llama-based models play a key role in storing knowledge of drug groups. Interestingly, the strongest causal effects are associated with intermediate tokens within the drug-group span, rather than the final token, contrary to prior findings on general factual knowledge in LLMs Meng et al. (2023). In addition, linear probing of drug-group activations reveals that pharmacological semantics are distributed across tokens rather than localized to individual positions, and that such semantics are already present in the embedding space. Pharmacological knowledge representation in LLMs remains understudied. This work provides a systematic investigation of how such knowledge is encoded and retrieved, establishing a foundation for mechanistic understanding of biomedical LLMs.

# 2 METHODS

## 2.1 DATASET CONSTRUCTION FOR BENCHMARKING AND ACTIVATION PATCHING

Although many benchmarks evaluate large language models (LLMs) on biomedical tasks, existing benchmarks do not explicitly target drug[1]-class relationships. We therefore constructed a dataset by parsing drug names from pharmacological actions categories curated by the U.S. National Library of Medicine National Center for Biotechnology Information. We excluded categories not directly related to pharmacological mechanisms (e.g. aerosol propellants). Using a curated dictionary of drugs and drug groups, we next generated a dataset for evaluation and activation patching experiments. We used two-choice question–answering dataset for two reasons. First, drug names are differently tokenized, so the first or last tokens may not preserve sufficient information to reliably identify the drug, making token-level evaluation impractical. Second, the correct answer is not unique, as multiple drugs may belong to the same pharmacological class. Therefore, we reformulated each query as a two-choice question by randomly sampling distractor options and varying the position of the correct answer. Predictions were obtained by selecting the answer choice with the maximum logit value among the two options. Model performance was evaluated using accuracy, as the dataset is balanced.

## 2.2 ACTIVATION PATCHING

The goal of this work is to investigate where the knowledge of drug classes is stored and if we can locate it in modern LLMs. To this end, we employed activation patching, a method widely used in mechanistic interpretability research to identify the specific model components responsible for particular functions and knowledge storage Meng et al. (2023). The method aims to causally identify activations that affect the model's output. Concretely, it consists of three forward passes

---

[1]By "drugs" we refer to all compounds listed in the U.S. National Library of Medicine National Center for Biotechnology Information, including approved drugs, compounds with known biological activity, and research compounds.

through the model: (1) a pass on a clean prompt, during which latent activations are cached, (2) a pass on a corrupted prompt, and (3) a pass on the corrupted prompt in which the activation of a selected model component is replaced with its corresponding activation from the clean cache. Specifically, we define a clean prompt with a known correct answer, for example:

*Question: Which compound is categorized as vasoconstrictor agents?*
*A) ergotamine*
*B) araldite*
*Answer:*

where the correct answer is A. We then construct a corresponding counterfactual prompt in which the different drug group is chosen such that the correct answer becomes B:

*Question: Which compound is categorized as bronchoconstrictor agents?*
*A) ergotamine*
*B) araldite*
*Answer:*

We performed activation patching by replacing selected activations from the counterfactual run with those from the clean run. We utilized symmetric token replacement in all experiments, as Gaussian noise may impair internal model mechanisms by introducing out-of-distribution inputs Zhang & Nanda (2023).

To evaluate the patching effect, we used a normalized logit difference metric, originally introduced for the indirect object identification (IOI) task Wang et al. (2022), defined as

$$\text{metric}(r, r') = \frac{\text{LD}_{\text{pt}}(r, r') - \text{LD}_*(r, r')}{\text{LD}_{\text{cl}}(r, r') - \text{LD}_*(r, r')}$$

where LD denotes the logit difference between the correct and incorrect answers, and the subscripts cl, $*$, and pt correspond to the clean, counterfactual, and patched runs, respectively. Model layers are indexed from 0 to 31. All patching experiments were performed using Transformer Lens package Nanda & Bloom (2022).

## 2.3 LINEAR PROBING

To test whether MLP modules in Llama-based models are critical for semantic representation, we constructed custom datasets containing paired drug groups with opposing semantic meanings. As a first group, we selected adrenergic receptor alpha-agonists and alpha-antagonists: agonists enhance the activity of a biological target, whereas antagonists inhibit it. To ensure that any observed effects reflect genuine semantic structure rather than token-specific artifacts, we included a second group consisting of central nervous system stimulants and depressants.

Specifically, we used a fixed two-answer prompt structure while varying the position of the drug group within the prompt. For each drug group (e.g., adrenergic alpha-agonists), we generated 300 prompts varying the position of the drug group in the prompt. Example prompts are shown in the Table 1.

Models were run on the full prompts, after which activations were subsetted to include only the token span corresponding to the drug group. We trained linear classifiers both on activations from individual tokens and on sum-pooled activations aggregated across all tokens within the span. To prevent data leakage, we used StratifiedKFold for sum-pooled tokens and StratifiedGroupKFold for individual tokens. Specifically, we used a logistic regression classifier from scikit-learn library Pedregosa et al. (2011) with regularization parameter C = $10^{-3}$ to avoid overfitting.

## 3 RESULTS

### 3.1 BIOMEDICAL AND GENERAL-PURPOSE LLMS ENCODE DRUG CLASS–NAME RELATIONSHIPS

We first assessed whether the models encode knowledge of drug class–name relationships by measuring the performance of several LLMs on clean prompts from the custom dataset described

Table 1: Example paired prompts used to probe semantic representations

| Example prompt for adrenergic alpha-agonists | Paired example prompt for adrenergic alpha-antagonists |
|---|---|
| Question: Which compound belongs to the class of adrenergic alpha-agonists?
A) mh-76 compound
B) ketoprofen
Answer: | Question: Which compound is known to act as adrenergic alpha-antagonists?
A) ketoconazole
B) quinidine
Answer: |
| **Example prompt for central nervous system depressants** | **Paired example prompt for central nervous system stimulants** |
| Question: Which compound would be grouped into central nervous system depressants?
A) tadalafil
B) amitriptyline, perphenazine drug combination
Answer: | Question: Which compound falls under central nervous system stimulants?
A) viomycin
B) 4-fluoroamphetamine
Answer: |

above. We considered both models fine-tuned on biomedical corpora and general-purpose LLMs, as biomedical fine-tuned models sometimes under-perform compared to general-purpose models on unseen tasks Dorfner et al. (2025). With the exception of BioGPT, all models demonstrated strong performance, with Llama-based models AI@Meta (2024) achieving superior accuracy. These results suggest that, with the exception of BioGPT, the models possess substantial knowledge of drug class–name associations (Table 2).

Table 2: Accuracy of models evaluated on the drug class–name relationship dataset.

| Model | Accuracy |
|---|---|
| BioGPT | 0.600 |
| OpenBioLLM-8B | **0.920** |
| BioMistral-7B | 0.860 |
| Llama-3.1-8B-Instruct | **0.900** |
| Gemma3-4B | 0.860 |

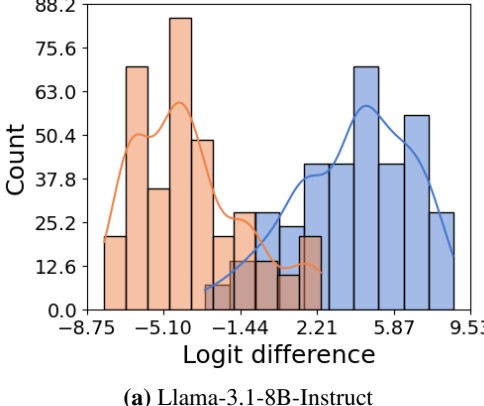

**(a)** Llama-3.1-8B-Instruct

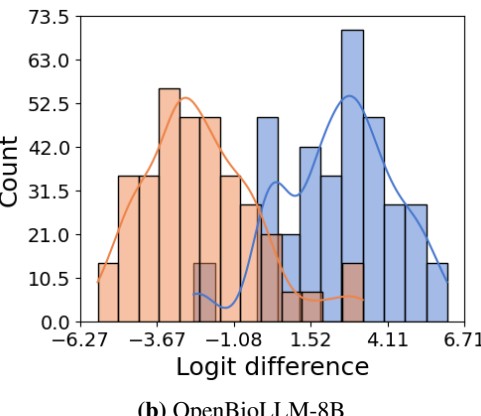

**(b)** OpenBioLLM-8B

Figure 1: Logit difference distributions for clean (blue) and counterfactual (orange) prompts across Llama-based models.

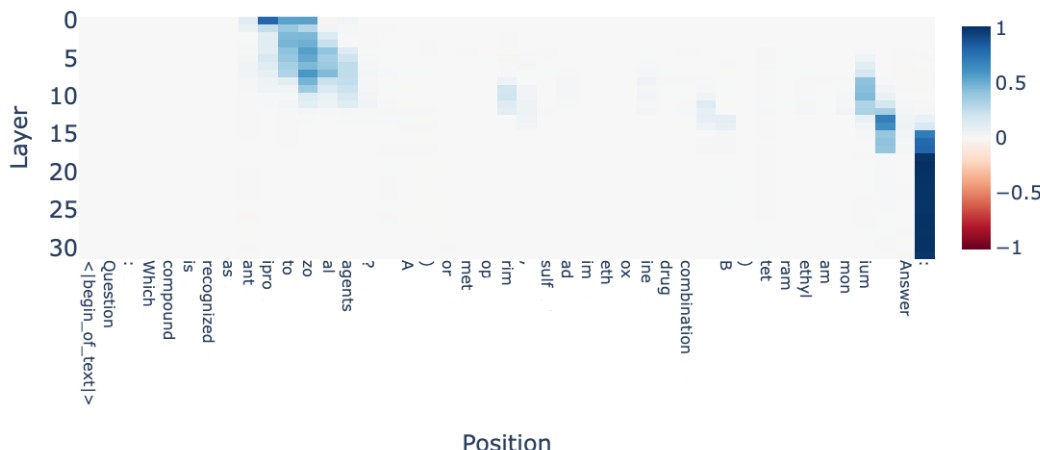

Figure 2: Activation patching of the Llama-3.1-8B-Instruct on the random prompt

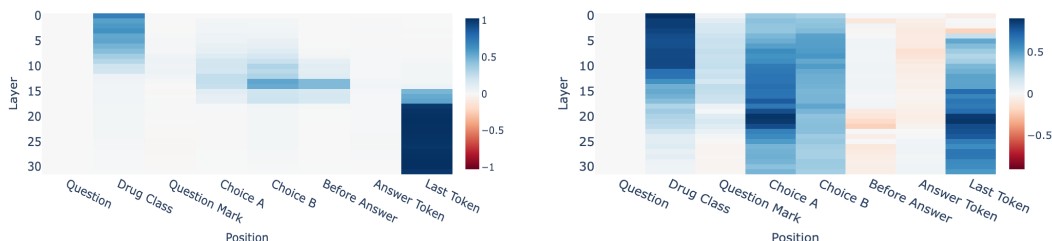

Figure 3: Activation patching of the residual stream of Llama-3.1-8B-Instruct (left) and OpenBioLLM-8B (right) across all prompts.

We additionally measured the logit difference for both clean and counterfactual prompts across Llama-based models. As expected, logit differences for clean prompts were distributed across positive values indicating a preference for the correct option. Conversely, for counterfactual prompts, in which the correct answer is reversed, the logit differences were predominantly negative.

## 3.2    ACTIVATION PATCHING OF THE RESIDUAL STREAM OF LLAMA-BASED MODELS

To investigate causal mechanisms, we performed activation patching of the residual stream of the OpenBioLLM-8B Saama AI Labs (2024) and Llama-3.1-8B-Instruct Meta AI (2024) at the start of each transformer block while accounting for token position. We observed a causal effect when patching drug group tokens at first ten layers. This indicates that task-relevant information injected at early layers can be propagated through the model and subsequently transformed by downstream layers to support task performance.

In contrast to prior work Meng et al. (2023), which reports that factual knowledge is mediated by strongly causal states at the final subject token, we find that in two-choice pharmacology prompts the maximal causal effect occurs at intermediate tokens within the drug group span. We speculate that this difference may reflect distinct circuit-level mechanisms underlying the formation of drug-group representations versus factual knowledge.

## 3.3    ACTIVATION PATCHING OF THE MLP LAYERS OF LLAMA-BASED MODELS

We also applied activation patching to MLP outputs as prior work Meng et al. (2023) showed that factual knowledge is stored in MLPs. We intervened on a contiguous window of MLP layers rather than a single layer, as restoring single activations from individual MLP had minimal effect. We

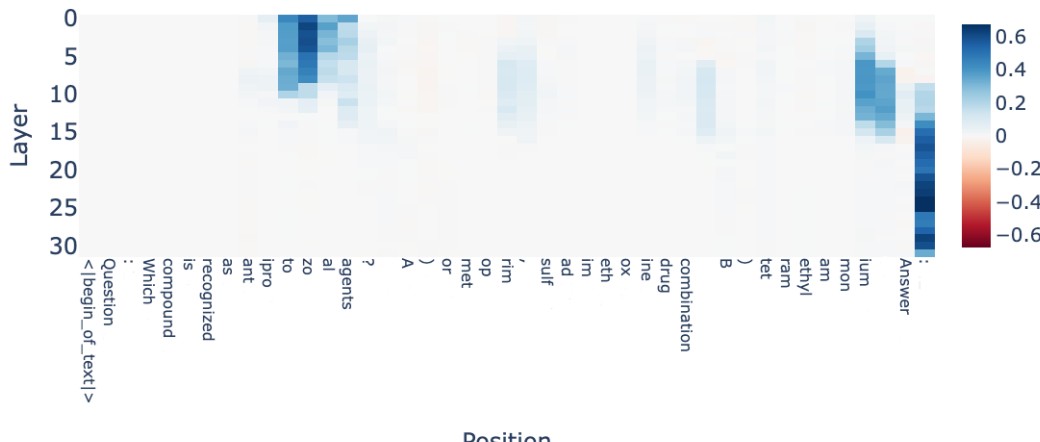

Figure 4: Activation patching of the Llama-3.1-8B-Instruct on the random prompt

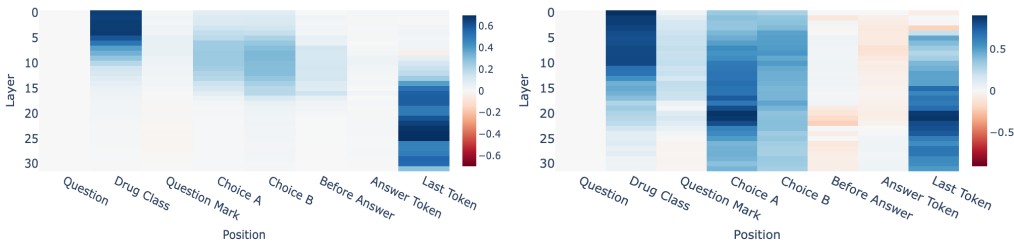

Figure 5: Activation patching of the MLPs of Llama-3.1-8B-Instruct (left) and OpenBioLLM-8B (right) across all prompts

applied a 10-layer window (5 layers before and 5 layers after the central layer, where available). Patching early MLP layers (layers 0-10) within the drug group span consistently produced positive effects. Similar to residual-stream patching, interventions on intermediate drug-group tokens exhibited strong causal impacts that were much larger than those of the final drug-group token. While interventions on the final token of the prompt are known to produce strong causal effects, the average maximum effect for drug-group tokens was 0.76, compared to 0.80 for the final token of the prompt, indicating that drug-group representations exert a substantial causal influence.

## 3.4 SEMANTIC REPRESENTATIONS ARE DISTRIBUTED ACROSS TOKENS FOR LLAMA-BASED MODELS

We identified a subset of early MLP layers with strong causal influence on model outputs. What functional role do these layers play? While attention blocks integrate information across the sequence, MLP blocks perform non-linear feature transformations that encode and refine semantic representations within individual token states. If semantic information is explicitly encoded in a token's activation, then a linear probe should be able to predict semantic labels directly from that activation.

To test this, we trained logistic regression on activations extracted from full prompts, separately for each layer, and subsetted to the drug-group span (see Methods for details). Although sum-pooled linear probes achieved near-perfect separability, probes trained on individual tokens exhibited substantially lower performance (Figure 6). This suggests that semantic class information is not aligned with single-token activations, but instead emerges through aggregation.

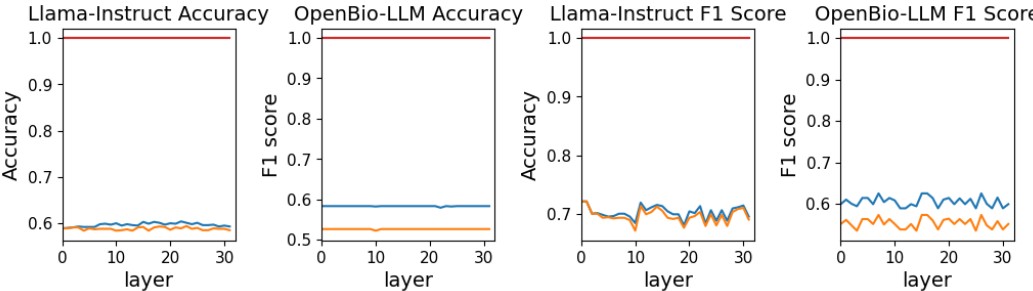

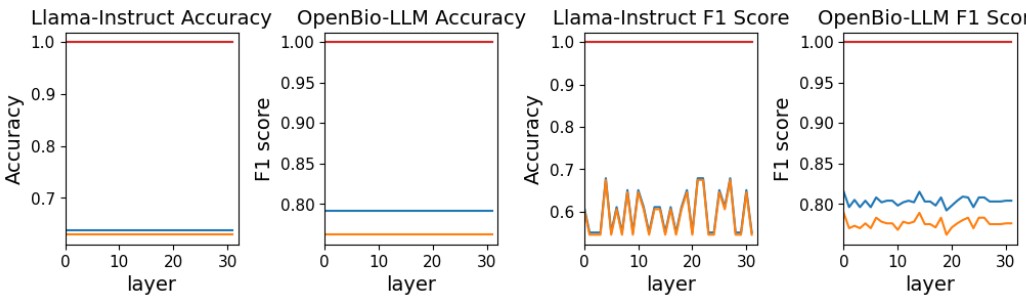

Figure 6: Metrics of linear probes trained on activations extracted from Llama-based models. Blue and orange lines indicate train and test metrics for the model trained on individual token activations, while red line shows both train and test performance for the model trained on sum-pooled tokens.

Given the similar performance observed across layers, we next asked whether semantic information is already present in the embedding space. To test this, we extracted activations from the residual stream prior to layer 0 (Supplementary Tables 3-10). Linear probes trained on sum-pooled activations achieved maximal performance for both drug-group classes and for both models, indicating that semantic information is already encoded at the embedding level. These results highlight similarities in drug-group representation spaces between instruction-fine-tuned and biomedical fine-tuned models.

## 3.5 PRIOR WORK

### 3.5.1 MECHANISTIC INTERPRETABILITY

Mechanistic interpretability aims to understand the internal computations of neural networks by characterizing where information is stored, how it is transformed across layers, and which internal components are causally responsible for specific outputs. Activation patching relies on causal interventions on model activations and have been used to localize factual knowledge Meng et al. (2023), induction circuits Conmy et al. (2023), and study implicit reasoning Li et al. (2024).

Complementary to causal methods, probing-based analyses evaluate whether specific information is decodable from internal representations. Probing-based analyses have been widely used to study the emergence of syntactic Hewitt & Manning (2019); Tenney et al. (2019), semantic Vulic et al. (2020); Richardson et al. (2019), and conceptual information Geva et al. (2022) across layers.

Together, causal patching and probing provide complementary views of internal representations: patching tests necessity and sufficiency, while linear probing assesses linear separation. In this work, we adopt both perspectives to study how pharmacological knowledge is encoded in biomedical LLMs, enabling a more mechanistic characterization of where drug-group semantics emerge, how they are distributed across tokens and layers.

### 3.5.2 MECHANISTIC INTERPRETABILITY FOR BIOMEDICAL LLMs

Despite its clinical significance, mechanistic interpretability in the biomedical domain remains underexplored, with most existing work focusing on general-domain LLMs and task-specific analyses. A prior study examines mechanistic interpretability in biomedical LLMs, using gender and race as attributes to probe internal representations Ahsan et al. (2025). That work finds that gender-related information is highly localized in intermediate MLP layers and can be reliably manipulated at inference time via activation patching, enabling targeted modifications of generated clinical vignettes and associated downstream clinical predictions. In contrast, representations associated with patient race appear more distributed, though still partially amenable to causal intervention. These findings motivate broader mechanistic investigations of biomedical LLMs that extend beyond specific attributes to a wider range of biomedical concepts.

## 4 LIMITATIONS

A limitation of this study is that our evaluation is restricted to drug groups. Future work is needed to examine the applicability of our approach to individual drugs and other biomedical categories. In addition, we do not explicitly analyze how drug-group concepts are composed from individual tokens or identify the specific attention heads or circuits responsible for their formation.

## 5 CONCLUSIONS

In this work, we provide a mechanistic analysis of how drug-group semantics are represented in biomedical large language models. Using a combination of causal activation patching and probing analyses, we show that pharmacological information is encoded early in the network, distributed across tokens, and not confined to final-token representations. Causal interventions reveal that intermediate tokens in early layers drive the strongest effects, while probing results demonstrate that pooled representations substantially outperform token-level probes and are linearly separable even before the first transformer layer. Together, these findings highlight the value of combining causal and correlational interpretability methods and advance a more transparent, mechanistic understanding of pharmacological knowledge in biomedical LLMs.

## 6 LLM USAGE STATEMENT

In accordance with ICLR 2026's policies on large language model usage, we disclose that we used a large language model as a general-purpose assistive tool to polish text and improve clarity in the writing of this manuscript. We did not use it to generate research hypotheses, conceptual ideas, interpretations, or technical content of the research. All substantive content, conceptual decisions, and scientific claims reported in this work are the authors' own, and we take full responsibility for them.

## 7 ACKNOWLEDGMENT

We are grateful to Akhil Jalan for his contributions during the early stages of this work, including organizing weekly meetings and providing helpful tutorials and feedback. We also thank Javier Ferrando and Ryan Lagasse for reviewing the paper prior to the workshop submission and for their constructive feedback. We further acknowledge the Algoverse organization for supporting and facilitating AI research efforts related to this work. Finally, we thank the Lambda team for providing GPU resources that enabled us to conduct the computational experiments presented in this study.

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

# A    SUPPLEMENTARY TABLES

Table 3: Linear probe performance on adrenergic group activations extracted before the first transformer layer of Llama-Instruct and trained on individual tokens

| Test Accuracy | Train Accuracy | Test F1 score | Train F1 score | Test ROC-AUC | Train ROC-AUC |
|---|---|---|---|---|---|
| $0.5200 \pm 0.0116$ | $0.5318 \pm 0.0029$ | $0.5895 \pm 0.1949$ | $0.5949 \pm 0.1975$ | $0.5556 \pm 0.0000$ | $0.5556 \pm 0.0000$ |

Table 4: Linear probe performance on adrenergic group activations extracted before the first transformer layer of Llama-Instruct and trained on sum-pooled tokens

| Test Accuracy | Train Accuracy | Test F1 score | Train F1 score | Test ROC-AUC | Train ROC-AUC |
|---|---|---|---|---|---|
| $1.0000 \pm 0.0000$ | $1.0000 \pm 0.0000$ | $1.0000 \pm 0.0000$ | $1.0000 \pm 0.0000$ | $1.0000 \pm 0.0000$ | $1.0000 \pm 0.0000$ |

Table 5: Linear probe performance on central nervous system group activations extracted before the first transformer layer of Llama-Instruct and trained on individual tokens

| Test Accuracy | Train Accuracy | Test F1 score | Train F1 score | Test ROC-AUC | Train ROC-AUC |
|---|---|---|---|---|---|
| $0.6291 \pm 0.0090$ | $0.6382 \pm 0.0022$ | $0.6962 \pm 0.0984$ | $0.7009 \pm 0.1005$ | $0.7333 \pm 0.0000$ | $0.7333 \pm 0.0000$ |

Table 6: Linear probe performance on central nervous system group activations extracted before the first transformer layer of Llama-Instruct and trained on sum-pooled tokens

| Test Accuracy | Train Accuracy | Test F1 score | Train F1 score | Test ROC-AUC | Train ROC-AUC |
|---|---|---|---|---|---|
| $1.0000 \pm 0.0000$ | $1.0000 \pm 0.0000$ | $1.0000 \pm 0.0000$ | $1.0000 \pm 0.0000$ | $1.0000 \pm 0.0000$ | $1.0000 \pm 0.0000$ |

Table 7: Linear probe performance on adrenergic group activations extracted before the first transformer layer of OpenBioLLM and trained on individual tokens

| Test Accuracy | Train Accuracy | Test F1 score | Train F1 score | Test ROC-AUC | Train ROC-AUC |
|---|---|---|---|---|---|
| $0.5521 \pm 0.0093$ | $0.5907 \pm 0.0093$ | $0.5388 \pm 0.0582$ | $0.5857 \pm 0.0669$ | $0.6111 \pm 0.0162$ | $0.6539 \pm 0.0156$ |

Table 8: Linear probe performance on adrenergic group activations extracted before the first transformer layer of OpenBioLLM and trained on sum-pooled tokens

| Test Accuracy | Train Accuracy | Test F1 score | Train F1 score | Test ROC-AUC | Train ROC-AUC |
|---|---|---|---|---|---|
| $1.0000 \pm 0.0000$ | $1.0000 \pm 0.0000$ | $1.0000 \pm 0.0000$ | $1.0000 \pm 0.0000$ | $1.0000 \pm 0.0000$ | $1.0000 \pm 0.0000$ |

Table 9: Linear probe performance on central nervous system group activations extracted before the first transformer layer of OpenBioLLM and trained on individual tokens

| Test Accuracy | Train Accuracy | Test F1 score | Train F1 score | Test ROC-AUC | Train ROC-AUC |
|---|---|---|---|---|---|
| $0.6371 \pm 0.0071$ | $0.6679 \pm 0.0079$ | $0.6254 \pm 0.0035$ | $0.6627 \pm 0.0116$ | $0.7416 \pm 0.0083$ | $0.7690 \pm 0.0096$ |

Table 10: Linear probe performance on central nervous system group activations extracted before the first transformer layer of OpenBioLLM and trained on sum-pooled tokens

| Test Accuracy | Train Accuracy | Test F1 score | Train F1 score | Test ROC-AUC | Train ROC-AUC |
|---|---|---|---|---|---|
| $1.0000 \pm 0.0000$ | $1.0000 \pm 0.0000$ | $1.0000 \pm 0.0000$ | $1.0000 \pm 0.0000$ | $1.0000 \pm 0.0000$ | $1.0000 \pm 0.0000$ |

