# OpenReview forum: "Tracing Pharmacological Knowledge in Large Language Models"
_ICLR.cc/2026/Workshop/LMRL — ICLR 2026 Workshop LMRL Poster_

### Official Review · Reviewer_y1me · 2026-02-20
**Good Work**

**Rating:** 6
**Confidence:** 3

**Review:**

This paper investigates how pharmacological knowledge, specifically drug–group semantics, is encoded and retrieved within Llama-based large language models. Using a combination of activation patching and linear probing, the study provides a mechanistic analysis of where and how drug-group information is represented across layers and token positions. The authors construct a custom two-choice benchmark derived from curated pharmacological databases to isolate drug class–name relationships. Results show that early transformer layers play a dominant causal role, with the strongest effects arising from intermediate tokens rather than final tokens, diverging from prior findings on general factual knowledge. Linear probing suggests that pharmacological semantics are distributed across token spans and already present in the embedding space. Overall, the work aims to establish a foundation for mechanistic interpretability of biomedical knowledge in LLMs.

**Strengths**

1. Addresses an underexplored but important question by providing a mechanistic analysis of pharmacological knowledge in large language models.
2. Combines causal interpretability methods with probing-based analysis in a principled and well-motivated experimental design.
3. Uses carefully constructed datasets and counterfactual prompts that reduce confounds related to tokenization and answer ambiguity.

**Weaknesses**

1. The analysis is limited to Llama-based architectures, leaving open whether the findings generalize to other model families.
2. The benchmark focuses narrowly on drug-group classification and does not test broader pharmacological reasoning or downstream task performance.
3. Linear probing results are mainly descriptive, with limited exploration of nonlinear or compositional representations.
4. The causal effects are reported primarily at the residual stream level, without deeper analysis of attention heads or MLP subcomponents.
5. Practical implications for improving model training or reliability in biomedical applications are only briefly discussed.

---

### Official Review · Reviewer_iW6i · 2026-02-23

**Rating:** 7
**Confidence:** 3

**Review:**

This work focuses on characterizing how drug-group semantics are represented and retrieved in LLMs. For this purpose, the authors constructed a 2-choice QA dataset for their experiments like activation patching, linear probing etc. The experiments lead to interesting conclusions about localization of drug-group knowledge and that they are distributed across the tokens rather than being centered at one token.

Strengths:
1) The paper reuses the existing evaluation tools but applies them in a novel setting.
2) The experiments are comprehensive and lead to interesting hypotheses.
3) The paper is well-written and is easy to follow.

Weaknesses:
1) The authors perform experiments on only 1 task involving drug-group knowledge. While this can form hypotheses, it is not a substantive evidence for the claims.
2) The experiment methodology is unclear at some places. For example, in linear probing, the authors mention creating 300 prompts with varying drug-group position, but don't specify clearly what are the variations across prompts?

---

### Official Review · Reviewer_FJQx · 2026-02-24
**This paper investigates how pharmacological knowledge is encoded in Llama-based biomedical models using activation patching and linear probing. It concludes that drug-group knowledge is distributed across tokens and established in early layers, with sum-pooled representations providing the most diagnostic information.**

**Rating:** 5
**Confidence:** 3

**Review:**

Overall Assessment
While relevant to clinical AI transparency, the findings are largely expected for transformer architectures. The methodology applies standard interpretability tools to a new dataset without introducing new theoretical frameworks or discovering unique model behaviors. It functions more as a case study than a fundamental advancement.

Strengths
Addresses a meaningful gap in clinical AI transparency.
Uses causal activation patching rather than simple correlation.
Provides useful data on how sub-word tokenization affects semantic representation.

Weaknesses
Limited technical novelty; tools like Logit Lens are used without domain-specific adaptation.
Predictable conclusions regarding distributed features and early-layer emergence.
Lacks mechanistic depth (no identification of specific attention heads or circuits).
Probing success does not necessarily prove the functional utility of those representations.

---

### Meta-Review · Area_Chair_ggjX · 2026-02-28

**Recommendation:** Accept (Poster)
**Confidence:** 4

**Metareview:**

While I tend to agree with Reviewer FJQx that these results are unsurprising given what we know about transformers more generally, there is enough interest from the other reviewers that I think this is worth discussing at the workshop.

---

### Decision · Program_Chairs · 2026-03-02

**Decision:**

Accept (Poster)

**Comment:**

Please see the meta-review.